# An Outbreak of Highly Pathogenic Avian Influenza (H7N7) in Australia and the Potential for Novel Influenza A Viruses to Emerge

**DOI:** 10.3390/microorganisms9081639

**Published:** 2021-07-31

**Authors:** Andrew T. Bisset, Gerard F. Hoyne

**Affiliations:** 1School of Nursing, Midwifery, Health Sciences and Physiotherapy, Faculty of Medicine, Nursing and Health Sciences, University of Notre Dame Australia, Fremantle, WA 6160, Australia; gerard.hoyne@nd.edu.au; 2Institute for Health Research, University of Notre Dame Australia, Fremantle, WA 6160, Australia; 3Centre for Cell Therapy and Regenerative Medicine, School of Biomedical Sciences, The University of Western Australia, Nedlands, WA 6009, Australia; 4School of Medical and Health Sciences, Edith Cowan University, Joondalup, WA 6027, Australia

**Keywords:** H7N7, avian influenza, emu, zoonotic, outbreak, Victoria, Australia

## Abstract

In 2020, several geographically isolated farms in Victoria, Australia, experienced an outbreak of highly pathogenic avian influenza (HPAI) virus H7N7 and low pathogenic avian influenza (LPAI) viruses H5N2 and H7N6. Effective containment and control measures ensured the eradication of these viruses but the event culminated in substantial loss of livestock and significant economic impact. The avian HPAI H7N7 virus generally does not infect humans; however, evidence shows the ocular pathway presents a favourable tissue tropism for human infection. Through antigenic drift, mutations in the H7N7 viral genome may increase virulence and pathogenicity in humans. The Victorian outbreak also detected LPAI H7N6 in emus at a commercial farm. Novel influenza A viruses can emerge by mixing different viral strains in a host susceptible to avian and human influenza strains. Studies show that emus are susceptible to infections from a wide range of influenza viral subtypes, including H5N1 and the pandemic H1N1. The emu’s internal organs and tissues express abundant cell surface sialic acid receptors that favour the attachment of avian and human influenza viruses, increasing the potential for internal genetic reassortment and the emergence of novel influenza A viruses. This review summarises the historical context of H7N7 in Australia, considers the potential for increased virulence and pathogenesis through mutations and draws attention to the emu as potentially an unrecognised viral mixing vessel.

## 1. Introduction

The prevalence of zoonotic infectious diseases is a constant across many continents and are considered a threat to global health, security and economic growth [1]. With global attention focused on the COVID-19 pandemic, the potential for the unexpected emergence of new zoonoses remains high, particularly with an ever-increasing anthropogenic impact of human development, environmental change and destruction of animal habitat. The emergence of a novel viral pathogen from an animal reservoir could have a significant economic and agricultural impact, but if the viral pathogen were to acquire the capacity for human transmission, then it could be responsible for another global pandemic, as observed with the H1N1 influenza virus in 1918 and SARS CoV2 virus in 2020.

Historically, the emergence of zoonotic infectious diseases within Australia is limited due to a combination of strict biosecurity measures, a willingness to report suspected outbreaks and the rapid containment thereof. Yet, outbreaks still occur despite these measures, often resulting in substantial economic loss and, rarely, loss of life. As an example, a novel equine virus (Hendra virus) emerged in the Brisbane suburb of Hendra Queensland in September 1994 [2]. The virus emerged as a spill-over from the Australian flying fox (*Pteropus* spp.) into horses, with subsequent transmission into humans from close contact with the infected horses [3,4]. The net result was the infection of 18 horses and two humans, resulting in the death of 14 horses and one stable hand [5]. To further highlight that zoonoses are not limited to land-based animals, tuberculosis (*Mycobacterium pinnipedii*) has been detected in an Australian Sea Lion [6].

Despite Australia’s effective biosecurity and containment measures, the greatest threat to the emergence of an unexpected infectious disease lies with the spread of a novel avian influenza virus (AIV), likely to be facilitated by the presence of intercontinental migratory birds within Australia [7,8,9]. The AIV is a type A influenza virus adapted to an avian host [10] with a primary reservoir found within aquatic bird species such as Anseriformes (ducks, geese, swans) and Charadriiformes (shorebirds, gulls, alcids) [11,12]. Transmission into domestic birds and poultry is seeded from within this reservoir, generally through the intermingling of species in farms or live poultry markets and through a lack of biosecurity that inadequately restricts the mixing of free-range and wild birds [13,14,15].

Even though Australia has experienced limited avian influenza outbreaks in domestic bird species, substantial economic loss is often attached to each event. While avian influenza viruses primarily infect avian hosts, they can also cross over into mammals, sometimes leading to infection and disease in humans. Of particular concern are the intermixing of avian and human influenza viral strains within cells of a susceptible host, leading to the so-called “mixing vessel” hypothesis [16]. The segmented genome of RNA viruses such as type A influenza can reassort with other influenza A viruses giving rise to a novel virus for which humans may have no pre-existing immunity.

The purpose of this review is to summarise a recent outbreak in Australia of the recurring influenza viral subtype H7N7 and its human zoonotic potential. Concurrent with this outbreak was the emergence of another influenza A viral strain (H7N6) in emus (*Dromaius novaehollandiae*), a large flightless bird native to Australia. The importance of this review is to emphasise the continuing circulation of H7N7 within the Australian mainland and highlight the emu as a bird susceptible to avian and human influenza viruses, with a tissue tropism favourable to viral reassortment that could potentially lead to the emergence of novel influenza A strains.

## 2. An Outbreak of Multiple Avian Influenza Virus Subtypes Simultaneously Emerge across Three Australian Farms

An outbreak of highly pathogenic AIV H7N7 occurred across three egg farms in Victoria, Australia, between 31 July and 25 August, 2020 (Table 1) [17]. These outbreaks are not uncommon in Australia, with recurring infections of both low pathogenic avian influenza (LPAI) and highly pathogenic avian influenza (HPAI) viruses reported since 1976 (Table 2) [18]. In each instance, outbreaks were contained and eradicated through effective quarantine and control measures, including this recent event [19].

One enigmatic aspect of the 2020 outbreak lies with the detection of three different IAV strains co-occurring across three bird species, which according to records, would be the first event in Australia where more than one viral subtype (LPAI or HPAI) has emerged simultaneously [18]. In addition to the detection of one HPAI virus (H7N7) in Victoria, two viral strains of LPAI were detected in turkeys (H5N2) and emus (H7N6). In this instance, the simultaneous emergence of three viral strains does not mean Australia is experiencing a rising risk of avian influenza outbreaks, nor does it indicate the presence of mutated strains with bird-to-human transmission capability. However, it highlights the likelihood that multiple avian influenza A viral subtypes are naturally circulating in domestic poultry, with viral transmission into domestic poultry likely to have been acquired through mixing with wild aquatic bird species [8].

While the co-detection of LPAI and HPAI viruses in a domestic poultry setting may be unexpected, the risk of pathogenic infections arising in the human population is considered minimal, primarily because of the host range restriction associated with all avian influenza viral subtypes [23]. However, random mutations in viral genomes can significantly alter the virulence and pathogenicity of a virus. The introduction of a pathogenic and virulent strain of avian influenza virus into the human population is rare, but when it happens, the pandemic potential increases if mutations confer efficient and effective human-to-human transmission [16]. Of the three avian influenza viral subtypes detected during the Victorian outbreak, only HPAI H7N7 has previously caused infections in humans [24,25,26]. Whilst HPAI H5Nx viruses are a known threat to the poultry industry and human health [27], there is no indication that HPAI H5 subtypes are present in Australia, although the risk of exposure from long-distance migratory birds is recognised [7]. However, influenza virsues are constantly evolving and the potential for HPAI H5N2 to emerge from an LPAI precursor has been demonstrated using ostrich-origin H5N2 progenitor viruses serially passaged in embryonated chicken eggs [28].

Recurring outbreaks of HPAI H7N7 in Australia are not indicative of a rising zoonosis risk from this strain. However, aquatic birds are an endemic reservoir that carries a genetically diverse range of influenza A viral subtypes that could infect birds, mammals and humans [11]. Moreover, influenza viruses are a constantly evolving pathogen with two primary mechanisms that promote change; antigenic drift and antigenic shift [29]. With the periodic but continuing resurgence of the HPAI H7N7 subtype in domestic poultry over the past 44 years, the emergence of a novel influenza virus may begin with antigenic drift.

## 3. Amino Acid Substitutions in the Avian Influenza A (H7N7) Genome Can Increase Virulence and Pathogenicity in Humans

The influenza virus replication cycle frequently introduces mutations into the viral genome due to the poor proof-reading processes of the virally encoded RNA polymerase [30]. Antigenic drift is caused by point mutations introduced into the viral genome, with a high rate of drift observed in antigenic regions of the hemagglutinin (HA) or neuraminidase (NA) glycoproteins compared to other viral genomic components. Single point mutations of an avian influenza viral genome can significantly affect virulence and pathogenicity in humans. For example, under experimental conditions, avian influenza virus A (H7N9) acquired a significant binding affinity for human trachea epithelial cells through the single point mutation HA-G228S [31]. Investigations into triple point mutations (V186G/K-K193T-G228S) of the H7 HA (H7N9 virus) have also demonstrated a lower binding affinity for avian α -2,3 sialic acid (SA) receptors and stronger binding for human α-2,6 SA receptors, raising the potential for human-to-human viral transmission [32]. It has also been shown that certain molecular substitutions in the viral polymerase basic protein 2 (PB2) of H7N9 can increase pathogenicity in animals [33]. The net result of viral genome mutations is the substitution of one or many amino acids into viral proteins with the potential for changes in virulence, pathogenicity and tropism. Even though the 2020 Australian outbreak of HPAI H7N7 was resolved, the potential remains for more virulent, pathogenic or infectious strains to evolve due to viral adaptations.

An example of one such event occurred in 2003 when domestic poultry farms in the Netherlands experienced a substantial outbreak of HPAI H7N7. Widespread detection of this pathogen was reported across 255 commercial poultry farms, with control measures destroying approximately 30 million chickens [24]. Aside from the economic devastation, this event correlated with a significant outbreak of avian influenza A virus (H7N7) in humans. Between 1 March and 16 May 2003, 89 people were infected with the H7N7 virus, 85 had conjunctivitis or influenza-like symptoms; however, influenza-like symptoms were less common in those testing positive for H7. A veterinarian involved in screening HPAI infected flocks developed acute conjunctivitis and later died due to acute pneumonia and related complications [34]. An H7N7 viral isolate (A/Netherlands/219/03) cultured from this patient differed by 14 amino acid substitutions compared to the first chicken isolate of the same virus (A/Chicken/Netherlands/1/03) [35], providing evidence of antigenic drift, which may have caused increased pathogenesis in humans. However, except for this one case, other human avian influenza virus isolates had few mutations. Despite the number of people exposed to AIV in this outbreak and several confirmed H7N7 infections, there was little evidence for sustained human-to-human transmission. Evidence of direct human transmission arose when secondary infections of influenza virus A subtype H7 were confirmed in three exposed contacts from two separate households where the occupants were living with either an infected poultry worker or farmer [24,36]. The lack of human-to-human transmission and prevalence of conjunctivitis supports the contention of viral attachment via the ocular epithelium [37].

The ocular tropism of influenza virus H7 subtypes in humans is linked to the dominance of α-2,3 SA receptors lining the corneal and conjunctival epithelia [38,39,40]. During the Netherlands outbreak, positive H7N7 cases presenting with conjunctivitis were greater than those displaying only influenza-like symptoms, supporting the contention that in this instance, viral entry into the human host was in part governed by the preferential binding of AIV H7N7 to α-2,3 linked SA residues on ocular tissue, albeit with low replication titres in the respiratory tract [35,37]. Once in the eye, viral transfer to the respiratory tissues is facilitated by the nasolacrimal system, particularly the nasolacrimal duct, which expresses both α-2,3 and α-2,6 linked SAs [39]. Whilst the tissue tropism of a respiratory virus is linked to the distribution of cellular receptors, a virus can only begin a successful respiratory infection once effective replication is achieved. A single point mutation in the polymerase basic 2 (PB2) subunit is one substitution known to achieve this.

Cross-species transmission of influenza viruses is mediated by the accumulation of mutations that allow adaptation to the new host. In particular, mutations in PB2 are an important factor in determining replication effectiveness. Within the PB2 subunit, the substitution of glutamic acid (E) for lysine (K) at residue 627 can induce efficient viral replication in mammalian hosts [41,42]. The importance and functionality of this specific mutation is underscored by its occurrence across many avian viral subtypes, including H5N1, H7N9, H4N6, H9N2, H5N8 and H7N7, and in each virus, this mutation signals the adaptation of an avian influenza virus to a mammalian host [41,43,44,45,46,47,48]. Consistent with this observation, sequencing of the H7N7 virus isolated from the Netherlands outbreak (A/Netherlands/219/03) confirmed the PB2-E627K substitution but found it absent in conjunctivitis only cases (A/Netherlands/33/03) [35]. In a later study, Jonges et al. (2014) asserted that the PB2-E627K mutation in the fatal case emerged following a single passage in a human in conjunction with the hemagglutinin mutation HA-K416R, although the significance of the latter was not determined [48]. Despite the commonality of the E627K mutation, the pathogenicity of the fatal case cannot be attributed entirely to this mutation. Significantly, de Witt et al. (2010) concluded that in this case, H7N7 pathogenicity is a polygenic trait with nearly all mutations that enhanced replication in human cells were already present in IAVs isolated from farm poultry [26]. The polygenic accumulation of known virulence and human adaptation markers was later confirmed to have occurred before transmission to humans, emphasising that HPAI viruses with pandemic potential can emerge directly from poultry without the requirement to adapt in the human host [48].

Other studies have demonstrated how specific amino acid substitutions have led to changes in virulence and pathogenicity of the H7N7 virus. While the studies supporting these changes are few, they reveal a plausible pathway in which antigenic drift could lead to the evolution of novel AIVs with increased virulence or pathogenicity [49,50,51,52,53,54]. Yet, the evidence that supports these changes occurring naturally is lacking, possibly a product of diminished surveillance for this viral subtype.

Early detection of LPAI or HPAI viruses and rapid enforcement of control measures is crucial in limiting the intra and inter-farm spread of disease. The 2020 Victorian outbreak of avian influenza viruses was unusual with three different strains of both LPAI and HPAI viruses detected simultaneously across three different bird species at geographically separated sites. While not an unprecedented event from a global perspective [55,56,57,58], it highlights the potential for asymptomatic circulation of multiple viral subtypes in domestic poultry and a native bird species. Aside from antigenic drift, the circulation and mixing of different viral subtypes within avian species promotes the evolution of novel influenza A viruses through genetic reassortment [59], albeit with a restricted host tropism. The evolution of avian influenza viruses capable of infecting humans often, but not always, stems from the genetic reassortment of avian and human viral genomes within a susceptible host. If such a host exists, the potential for genetic reassortment and the emergence of novel influenza A viruses is real.

## 4. The Emu (*Dromaius novaehollandiae*) Has Abundant Co-Expression of α-2,3 and α-2,6 Sialic Acid Receptors on Internal Tissue and Organs

The 2020 outbreak in Victoria recorded a LPAI H7N6 virus in emus (*Dromaius novaehollandiae*), a flightless bird native to Australia, that was successfully eradicated through appropriate biocontainment measures with no further registered events related to this outbreak [17]. The emu has received scant attention as a reservoir of avian influenza viruses and even less so regarding its potential as a viral mixing vessel. However, this bird appears to have an extensive internal tissue tropism that makes it susceptible to avian (H5N1, H5N2) and human (H1N1) influenza viruses [60,61]. Additionally, reports of seropositivity for hemagglutinin subtypes H1–H9, H11, H12 and neuraminidase subtypes N1–N9 highlights how susceptible this bird is to a range of viral subtypes [62,63]. Recognising that novel influenza A viruses emerge through reassortment of viral subtypes, it is appropriate to consider how these findings raise the spectre for internal mixing and viral reassortment within this species and whether this possibility necessitates a greater research focus to mitigate the emergence of strains with pandemic potential.

In the only study of its kind, Gujjar et al. (2017) highlight the extensive distribution of α-2,3 and α-2,6 SA receptors within multiple organs and tissues of the emu, using lectin histochemistry to distinguish between these two types of receptors (Table 3) [60]. The results showed broad expression of both SA receptor types across various organ and tissue samples within this species (Figure 1 and Figure 3). While this study reveals that many cell surfaces have one type of SA receptor, the immunoflourescent staining confirmed that specific cells within the bronchi (Figure 2) and spleen (Figure 3) co-express both types of receptors, especially for cells around the splenic capsule. A cell may constitutively express surface SA receptors for either avian or mammalian influenza A viruses (IAV); however, the emu appears to have tissue cells with specificity for both. If this is the case, that may indicate a host tropism potentially favourable to infection from avian and human IAVs with an undetermined gene reassortment capability.

The tissue tropism of avian influenza viruses restricts their infectivity to cells expressing α-2,3 SAs and human influenza viruses to cells expressing the α-2,6 SAs [64]. Changes in virulence, pathogenicity, and tropism may occur through antigenic drift, a process of random amino acid substitutions within the viral genome, resulting from the error-prone proof-reading replication mechanism [30]. More generally, not only do influenza viruses evolve by antigenic drift, novel viruses can emerge by reassorting viral RNA, and less frequently, non-homologous recombination [12]. It is the reassortment of viral RNA from different IAV subtypes that has the greatest potential to generate a novel influenza A virus, one for which there is potentially no pre-existing immunity in humans. With the emu displaying a cell tropism that could be favourable to both avian and human influenza A strains, should this species be recognised as a possible mixing vessel for the emergence of novel IAV strains? Evidence shows that the emu can carry human and avian IAVs [60] and many viral HA and NA subtypes [61,62,63,65,66], supporting the contention that these birds could provide the host conditions to facilitate genetic reassortment and give rise to novel IAV strains.

## 5. *Dromaius novaehollandiae* Is Susceptible to A Variety of Avian and Human Influenza A Viral Subtypes, Including the Pandemic H1N1

Gujjar et al. (2017) raise the possibility that genetic reassortment of avian and human IAVs could occur within the emu, although no evidence supports this contention. Nonetheless, the authors demonstrated SA receptor affinity in emus for human pandemic influenza H1N1 (A/H1N1/Virginia/2009) and LPAI H5N2 (A/chicken/PA/7659/85) viruses [60]. A study of captive birds from a zoo and two safari parks in Bangladesh concluded that in comparison to other birds in the same facilities, emus returned the highest seropositivity for AIV antibody prevalence (H5, H7, H9 subtypes) and AIV H9 subtype prevalence, thus highlighting the circulation of LPAI viruses in this specific setting [63]. In China, Kang et al. (2006) reported the pathogenic emergence of H9N2 in emus, postulating transmission from chickens as the likely origin of infection [67].

Leigh Perkins and Swayne (2002) investigated the susceptibility of emus to the highly pathogenic H5N1 virus, demonstrating how intranasal inoculation with A/chicken/Hong Kong/220/97 (H5N1) induced a high level of morbidity but no mortality [61]. In 2010, a naturally occurring outbreak of HPAI H5N1 was detected in emus at an Israeli mini-zoo at the Ein Gedi oasis near the Dead Sea [66]. Interestingly, the authors postulated migratory birds as the source of infection since, at the time, Israel was free from H5 infections, there was no commercial poultry within a 12 km radius, and no new birds had been introduced in the previous six months.

Whilst these results are minimally substantive in drawing attention to the susceptibility of emus to avian and human influenza A viruses, they do not validate the emu as a viral mixing vessel. Determining that function requires further research. Instead, the viral influenza infections reported in emus raise the spectre that should reassortment be possible; there is a significant unrealised potential for the emergence of novel influenza A viruses from within this species. Finally, although the emu is a bird native to Australia and commercially farmed there, this species is now also farmed in other parts of the world where viral subtypes not found in Australia are present.

## 6. Conclusions

Australia continues to experience periodic outbreaks of the H7N7 virus in commercial bird farms, managing and resolving each outbreak through effective biosecurity and control measures. The economic loss from these viral incursions is substantial due to the necessary remediation measures, often resulting in the widespread destruction of the infected birds. The HPAIV H7N7 virus generally poses no threat to the human population due to a binding specificity for avian α-2,3 SA receptors. However, the ocular tropism of this virus is evident from past outbreaks and serves as a reminder that in some circumstances, human eyes can serve as a portal of entry for AIVs.

Antigenic drift can alter the transmissibility, virulence and pathogenicity of a virus. Limited studies on H7N7 have demonstrated that changes in virulence and pathogenicity are possible through specific amino acid substitutions.

In conjunction with the detection of HPAI H7N7 at three Victorian egg farms was the simultaneous outbreak of LPAI H5N2 in turkeys and LPAI H7N6 in emus. The significance of these temporally related events is unknown, except to possibly conclude that the type and prevalence of avian influenza viruses circulating in commercial bird farms are unknown when asymptomatic.

Although the results of one study indicate emus have a favourable tissue tropism susceptible to infection from both avian and human influenza A viruses, it would be premature to accept those results as conclusive without further research. Despite this, there is sufficient evidence to corroborate that emus are susceptible to a range of influenza viral subtypes with a real possibility for genetic reassortment and the emergence of novel influenza A viruses.

## Figures and Tables

**Figure 1 microorganisms-09-01639-f001:**
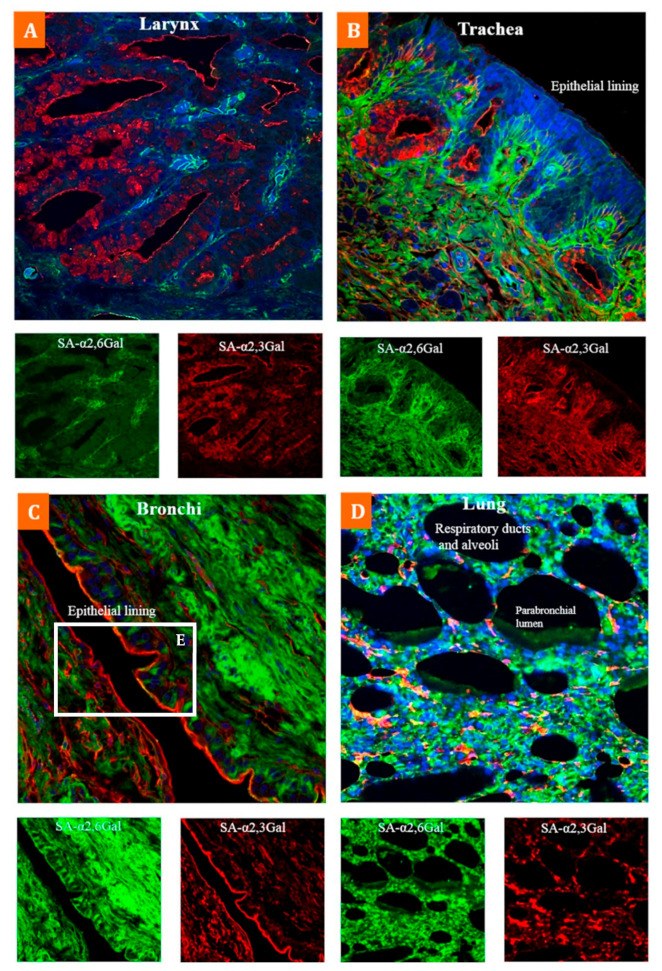
Co-expression of α-2,3 SA and α-2,6 SA receptors in the emu respiratory tract: Composite confocal images show an abundant co-expression of α-2,6 SA receptors (green) and α-2,3 SA receptors (red) throughout the emu respiratory tract. Comparable expression of both receptor types was observed in the ciliated epithelial cells, goblet cells and non-ciliated epithelial cells of (**A**) larynx, (**B**) trachea, (**C**) bronchi and (**D**) alveoli of lungs. Tissue sections stained with biotinylated MAAII (red-specific for α-2,3-Gal SA) and FITC labelled SNA (green-specific for α-2,6-Gal SA) lectins, and nuclear staining with DAPI (blue). Adapted with permission from ref. [60]. Copyright 2017 Elsevier.

**Figure 2 microorganisms-09-01639-f002:**
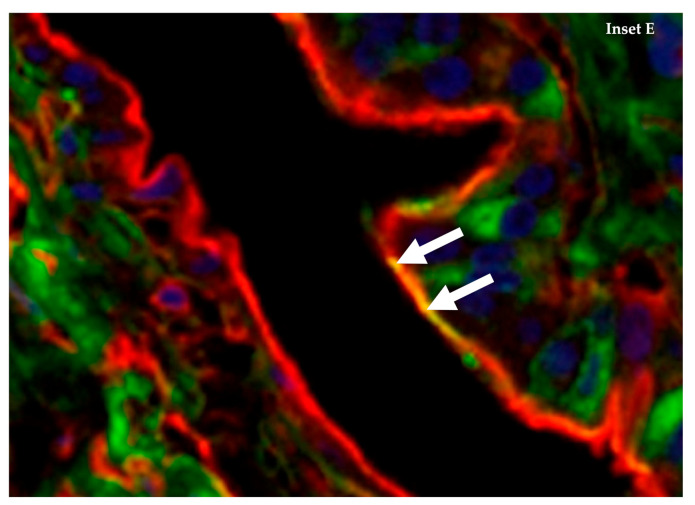
Close up view of Inset E from Figure 1 with arrows pointing to cells from the epithelial lining of the bronchi colour stained for both α-2,3 SA and α-2,6 SA receptors (yellow indicates a mixture of red and green staining). Tissue sections stained with biotinylated MAAII (red-specific for α-2,3-Gal SA) and FITC labelled SNA (green-specific for α-2,6-Gal SA) lectins, and nuclear staining with DAPI (blue). Adapted with permission from ref. [60]. Copyright 2017 Elsevier.

**Figure 3 microorganisms-09-01639-f003:**
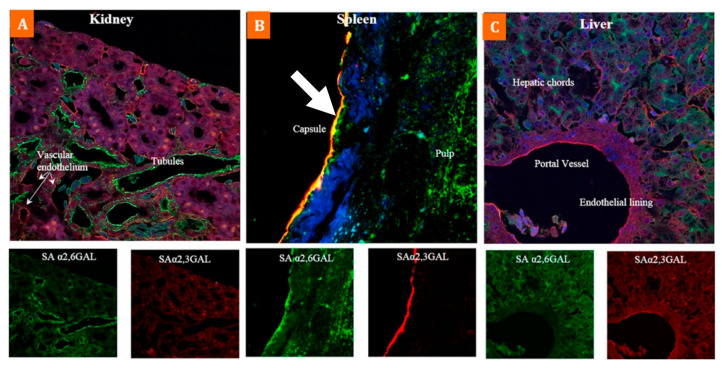
Abundant co-expression of α-2,3 SA and α-2,6 SA receptors in internal organs of the emu for (**A**) kidney, (**B**) spleen and (**C**) liver. Substantial yellow colouring of the spleen capsule lining indicates an abundance of cells with both types of SA receptors (white arrow). Tissue sections stained with biotinylated MAAII (red-specific for α-2,3-Gal SA) and FITC labelled SNA (green-specific for α-2,6-Gal SA) lectins, and nuclear staining with DAPI (blue). Adapted with permission from ref. [60]. Copyright 2017 Elsevier.

**Table 1 microorganisms-09-01639-t001:** Avian influenza A viruses detected during an outbreak in Victoria, Australia, between 31 July and 25 August, 2020.

Avian Influenza Virus	Detected in	Location (Farm Numbers)	Date of Last Reported Occurrence
H7N7 (HPAI)	Chicken	Lethbridge, Victoria (3)	21 February 2014
H5N2 (LPAI)	Turkey	Lethbridge, Victoria (1) Bairnsdale, Victoria (1)	26 June 2013
H7N6 (LPAI)	Emu	Kerang, Victoria (1)	2007

**Table 2 microorganisms-09-01639-t002:** Historical outbreaks of HPAI H7 subtypes in Australia since 1976.

Year	Avian Influenza Subtype	Description of Outbreak
1976	H7N7 (LPAI, HPAI)	In a combined broiler and egg farm in the outer suburbs of Melbourne, HPAI A/chicken/Victoria/76 was isolated. LPAI H7 isolated from a separate duck farm (A/duck/Victoria/76).
1985	H7N7 (HPAI)	Occurred in a chicken farm near Bendigo, Victoria H7N7 (A/chicken/Victoria/85) was isolated. Tests showed this virus to be highly virulent and not substantially different from A/chicken/Victoria/76 [20].
1992	H7N3 (HPAI)	Chicken farm near Bendigo, Victoria (different location to the 1985 outbreak) H7N3 (A/chicken/Victoria/92/1) isolated and determined to be HPAI virus.
1994	H7N3 (HPAI)	Egg farm in Brisbane, Queensland H7N3 (A/chicken/Qld/94) was isolated but different to the H7N3 virus isolated from Victoria in 1992.
1997	H7N4 (HPAI)	H7N4 was isolated from two chicken broiler-breeder farms and one emu farm near Tamworth, NSW. Viruses isolated from the three properties all had an identical HA cleavage site amino acid sequence. This sequence was also identical to that of the virus identified from the 1994 Queensland outbreak. Control measures resulted in the destruction of 310,565 chickens, 1,232,074 chicken eggs, 261 emu chicks and 147 emu eggs [21].
2012	H7N7 (HPAI)	Free range egg laying farm in Maitland, Lower Hunter Valley, NSW. 45,000 birds destroyed as a control measure.
2013	H7N2 (HPAI)	Two properties infected with HPAI H7N2 with 100% correlation of the same virus isolated from each property. The source of infection at the first property was confirmed to be from wild birds. Control measures on both properties resulted in the destruction of 471,380 birds [17,22].
2020	H7N7 (HPAI) H5N2 (LPAI) H7N6 (LPAI)	An outbreak of three different viral strains of AIV across six different farms in Victoria. Three egg farms (HPAI H7N7), two turkey farms (LPAI H5N2) and one emu farm (LPAI H7N6). OIE reports indicate more than 433,000 birds were destroyed as a control measure [17].

**Table 3 microorganisms-09-01639-t003:** Distribution of α-2,3 and α-2,6 sialic acid receptors within organs and tissues of *Dromaius novaehollandiae*.

Segment	Organ	Tissue	Qualitative Level of Sialic-Acid Receptor Expression
Respiratory Tract		Larynx-mucosa	α-2,3 + α-2,6
		Trachea-mucosa	α-2,3 + α-2,6
		Bronchi-mucosa	α-2,3 + α-2,6
		Alveoli-mucosa	α-2,3 + α-2,6
		Ciliated and non-ciliated epithelial cells, goblet cells	α-2,3 + α-2,6
		Submucosa	Higher α-2,6
Digestive Tract	Proventriculus-Duodenum	Epithelial cells	α-2,6 dominant throughout
	Lower portion of small intestine	Some parts of epithelial cells and mainly goblet cells	α-2,3 confined
	Large intestine	Epithelial lining of villi and goblet cells	Predominantly α-2,3
		Luminal region of villi	α-2,6 weakly expressed, concentrated here
	Duodenum-Colon		Increased expression of α-2,3
	Caecal tonsil Lymphoid organ		Abundant expression of α-2,3 + α-2,6
Other Major Organs	Kidney	Vascular endothelial wall	α-2,3 + α-2,6
		Tubules	α-2,6
	Liver		α-2,3 predominantly, especially endothelium of the portal vein
	Spleen	Capsule	α-2,3 + α-2,6 (uniform)
		Pulp region	Weak α-2,6
	Skin	Endothelial lining of veins	α-2,3 > α-2,6
	Brain		Consistent but sparse distribution of both receptors
	Skeletal Muscle	Around nuclei	α-2,3 + α-2,6 weakly present
		Basement membrane of muscle fibres	α-2,3 weakly present
	Heart		α-2,3 + α-2,6
		Blood capillaries	α-2,3 higher expression

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
