# Peer review of "An Outbreak of Highly Pathogenic Avian Influenza (H7N7) in Australia and the Potential for Novel Influenza A Viruses to Emerge"

_microorganisms, 2021, doi:10.3390/microorganisms9081639_

Round 1

Reviewer 1 Report

The authors summarized the findings of high and low pathogenic avian influenza occurred in Australia, which possess an unique feature compared with the outbreaks in other continents. Based on the findings, circulation of LPAIV, especially H7N7 subtypes is probable in this continent, but cause very low risk of zoonotic infection. However, given the fact that emu may play a role of “mixing vessel” as well as three different subtypes are prevalent in a very close distance, public health concern of emerging human-to-human infection pathogens should be steadily arising. Overall, the authors summarized the scientific evidence with clear purposes, which is so attractive to audience within and without the Australian continent. The authors need several minor revisions to make this review article published.

Line 113: Remove “(Wille 2019)”

Line 118-120: It is hard to understand this sentence. Increase of AIV pathogenicity during circulation among poultry is commonly known. However this sentence looks like to be against this knowledge. Please clarify the point of this sentence.

Line 184: Reference should be [42], not [41] in this content.

Reviewer 2 Report

The proposed article "An Outbreak of Highly Pathogenic Avian Influenza (H7N7) in Australia and the Potential for Novel Influenza A Viruses to Emerge" is a complete and complete analysis of the latest outbreaks of the influenza virus on the Australian continent. The article is of undoubted interest for specialists studying the influenza virus and for veterinarians. It should be especially noted that it is devoted to the emergence and spread of unique variants of avian influenza on such an isolated continent as"Australia". I believe that the information can be published. There are no comments to the authors.
